# IGF1R Deficiency Modulates Brain Signaling Pathways and Disturbs Mitochondria and Redox Homeostasis

**DOI:** 10.3390/biomedicines9020158

**Published:** 2021-02-06

**Authors:** Susana Cardoso, Icíar P. López, Sergio Piñeiro-Hermida, José G. Pichel, Paula I. Moreira

**Affiliations:** 1CNC—Center for Neuroscience and Cell Biology, University of Coimbra, 3004–504 Coimbra, Portugal; 2IIIUC—Institute for Interdisciplinary Research, University of Coimbra, 3030-789 Coimbra, Portugal; 3CIBB—Center for Innovative Biomedicine and Biotechnology, University of Coimbra, 3004-504 Coimbra, Portugal; 4Lung Cancer and Respiratory Diseases Unit (CIBIR), Fundación Rioja Salud, 26006 Logroño, Spain; iplgarcia@riojasalud.es (I.P.L.); jgpichel@riojasalud.es (J.G.P.); 5Telomeres and Telomerase Group, Molecular Oncology Program, Spanish National Cancer Centre (CNIO), 28029 Madrid, Spain; spineiro@cnio.es; 6Biomedical Research Networking Center in Respiratory Diseases (CIBERES), ISCIII, 28029 Madrid, Spain; 7Institute of Physiology, Faculty of Medicine, University of Coimbra, 3004-517 Coimbra, Portugal

**Keywords:** brain, insulin-like growth factor type 1 receptor, IGF1R-mediated signaling, mitochondria, redox balance, *UBC-CreERT2*, *Igf1r^fl/fl^*, IGF1R-deficient mice

## Abstract

Insulin-like growth factor 1 receptor (IGF1R)-mediated signaling pathways modulate important neurophysiological aspects in the central nervous system, including neurogenesis, synaptic plasticity and complex cognitive functions. In the present study, we intended to characterize the impact of IGF1R deficiency in the brain, focusing on PI3K/Akt and MAPK/ERK1/2 signaling pathways and mitochondria-related parameters. For this purpose, we used 13-week-old *UBC-CreERT2*; *Igf1r^fl/fl^* male mice in which *Igf1r* was conditionally deleted. IGF1R deficiency caused a decrease in brain weight as well as the activation of the IR/PI3K/Akt and inhibition of the MAPK/ERK1/2/CREB signaling pathways. Despite no alterations in the activity of caspases 3 and 9, a significant alteration in phosphorylated GSK3β and an increase in phosphorylated Tau protein levels were observed. In addition, significant disturbances in mitochondrial dynamics and content and altered activity of the mitochondrial respiratory chain complexes were noticed. An increase in oxidative stress, characterized by decreased nuclear factor E2-related factor 2 (NRF2) protein levels and aconitase activity and increased H_2_O_2_ levels were also found in the brain of IGF1R-deficient mice. Overall, our observations confirm the complexity of IGF1R in mediating brain signaling responses and suggest that its deficiency negatively impacts brain cells homeostasis and survival by affecting mitochondria and redox homeostasis.

## 1. Introduction

Insulin-like growth factor type 1 receptor (IGF1R) is an important cellular modulator with essential roles in the regulation of growth, development and metabolism, as well as in cellular processes like proliferation, survival, cell migration and differentiation; affecting nearly every organ system in the body [1,2,3]. Belonging to the family of transmembrane receptor tyrosine kinases (RTKs), where it is also included its highly homologous insulin receptor (IR), IGF1R is a transmembrane heterotetrameric glycoprotein consisting of two extracellular α subunits containing the IGF-binding site and two intracellular β subunits that exhibit tyrosine kinase activity [4]. Upon ligand-binding, IGF1Rs form constitutive homodimers that undergo ligand-dependent conformational changes in the α subunits and autophosphorylation of the β subunits, subsequently leading to the phosphorylation of insulin receptor substrates (IRS). This activity ensues activation of downstream intracellular signaling pathways: (1) the canonical phosphatidylinositol 3-kinase/protein kinase B (PI3K/Akt) and (2) the mitogen-activated protein (MAP) kinase (also called extracellular signal-regulated kinase; ERK), which ultimately transduce canonical ligand actions to control a number of essential biological outcomes [5,6,7].

Apart from its peripheral effects, IGF1R is present in different areas of the human brain like the hippocampus, prefrontal cortex, amygdala and parahippocampal gyrus, where its signaling fulfills important roles in neuronal development, neurogenesis and function, normal brain physiology and metabolism [8]. By activating downstream signaling cascades, IGF1/IGF1R is shown to participate in brain cell reparation, remodeling and resistance to stress during aging or in response to tissue damage for review see ref [1]. However, contradictions remain, and different transgenic animals have been developed to characterize and evaluate the involvement of the IGF1/IGF1R system in the central nervous system. While classical IGF1R knockout mice die at birth displaying generalized developmental abnormalities with multiorgan failure [9,10,11,12], the recently described *UBC-CreERT2*; *Igf1r^fl/fl^* mutant mice are apparently healthy upon the postnatal induction of tamoxifen-mediated *Igf1r* gene deletion. *Igf1r* deletion efficiently occurs in multiple mouse organs, making these mouse mutants a suitable animal model for the study of generalized IGF1R deficiency during postnatal development [13,14]. *UBC-CreERT2*; *Igf1r^fl/fl^* mutant males show diminished total body weight as well as altered organs weights than the respective controls (*Igf1r^fl/fl^*) when analyzed at eight weeks of age. In particular, at this age, the brain of these mutants was one of the few organs with a significantly lower weight compared to *Igf1r^fl/fl^* mice, a difference that faded out when corrected and expressed as organ-to-body weight ratio [13]. However, the consequences of postnatal IGF1R deficiency in brain tissue homeostasis in this particular animal model have not yet been investigated. In this framework, our main goal was to evaluate and dissect the consequences of IGF1R deficiency in the brain, putting the focus on PI3K/Akt and MAPK/ERK1/2 signaling pathways and mitochondria-related features. For this purpose, brain homogenates from 13-week-old *UBC-CreERT2*; *Igf1r^fl/fl^* mice and respective control mice were used to evaluate key proteins downstream of IGF1R signaling, synaptic integrity, caspases 3 and 9-like activity, mitochondrial dynamics, content and respiratory chain complexes activity, and general brain oxidative status. In brief, we have found that IGF1R deficiency distinctively modulates brain canonical signaling pathways, alters brain redox homeostasis and significantly impacts brain mitochondria content, structure and function, which ultimately will disturb brain homeostasis.

## 2. Experimental Section

### 2.1. Material and Reagents

All reagents used were of the highest analytical grade of purity commercially available and all aqueous solutions were prepared in ultrapure (type I) water. Amplex red, horseradish peroxidase type VI-A (HRP), antimycin A, 2,6-dichlorophenolindophenol (DCPIP), coenzyme Q1, bovine serum albumin (BSA), β-nicotinamide adenine dinucleotide (NADH), rotenone, decylubiquinol, n-dodecyl-β-d-maltoside, cytochrome c, 5–5-dithiobis(2-nitrobenzoic acid; DTNB), acetyl-CoA and oxaloacetate were provided by Sigma-Aldrich (St. Louis, MO, USA). Proteases and phosphatases inhibitors were obtained from Roche Applied Science (Amadora, Portugal). Caspase 3 and caspase 9 substrates were obtained from Merck (Lisboa, Portugal). Antibodies were obtained from the following sources: Akt (1:1000, BD Biosciences, San Jose, CA, USA (610861)); pAkt (Ser473) (1:1000, Cell Signaling, Danvers, MA, USA (4051)); CREB (1:750, Cell Signaling, Danvers, MA, USA (9197)); pCREB (Ser133) (1:1000, Millipore, Darmstadt, Germany (06-519)); DRP1 (1:1000, BD Biosciences, San Jose, CA, USA (611113)); pDRP1 (Ser616) (1:1000, Cell Signaling, Danvers, MA, USA (3455)); MAPK (ERK1/2) (1:1000, Cell Signaling, Danvers, MA, USA (9102)); pMAPK (ERK1/2) (Thr202/Tyr204) (1:1000, Cell Signaling, Danvers, MA, USA (4377)); pGSK3β (Ser9) (1:1000, Cell Signaling, Danvers, MA, USA (9336)); pGSK3β (Y216) (1:1000, Santa Cruz Biotechnology, Heidelberg, Germany (sc-135653)); GSK3β (1:1000, Santa Cruz Biotechnology, Heidelberg, Germany (sc-81462)); IGF-1R β(H60) (1:1000, Santa Cruz Biotechnology (sc-9038)); IR β(4B8) (1:1000, Cell Signaling, Danvers, MA, USA (3025)); ND1 (C-18) (1:1000, Santa Cruz Biotechnology, Heidelberg, Germany (sc-20493)); NRF2 (1:1000, Abcam, Cambridge, UK (ab31163)); NRF1 (1:1000, Santa Cruz Biotechnology, Heidelberg, Germany (sc-33771)); TFAM (1:750, Santa Cruz Biotechnology, Heidelberg, Germany (sc-23588)); Mfn1 (1:1000, Santa Cruz Biotechnology, Heidelberg, Germany (sc-50330)); Mfn2 (1:1000, Santa Cruz Biotechnology, Heidelberg, Germany (sc-100560)); MTCO1 (1:1000, Abcam, Cambridge, UK (ab14705)); OPA1 (1:1000, BD Biosciences, San Jose, CA, USA (612607)); PI3K (p110-alpha-C73F8) (1:1000, Cell Signaling, Danvers, MA, USA (4249S)); PSD95 (1:1000, Cell Signaling, Danvers, MA, USA (D27E11)); synaptophysin (1:1000, Sigma-Aldrich, St. Louis, MO, USA (S5768)); SNAP25 (1:1000, Sigma-Aldrich, St. Louis, MO, USA (S5187)); pTau (Ser 396) (1:1000, Santa Cruz Biotechnology, Heidelberg, Germany (sc-101815)); Tau (1:750, Sigma-Aldrich, St. Louis, MO, USA (T9450); actin (1:5000, Sigma-Aldrich, St. Louis, MO, USA (A5441)).

### 2.2. Animals

The present work was performed in brain tissue homogenates from 13-week-old male *UBC-CreERT2*; *Igf1r^fl/fl^* and *Igf1r^fl/fl^* mutant mice [13]. Brains were dissected from 5 h-fasted animals (6:00 to 11:00 a.m.) before sacrifice, snap-frozen in liquid nitrogen and kept at −80 °C until further analysis. As described, these transgenic mice were generated by mating *UBC-CreERT2* transgenic with *Igf1r^fl/fl^* mutant mice. Briefly, *UBC-CreERT2* mice express a Cre-ERT2 fusion protein composed of Cre recombinase and a mutant form of the estrogen receptor with a triple mutation, which is selectively activated in the presence of tamoxifen. *Igf1r^fl/fl^* mice contain loxP sites flanking exon 3 of the IGF1R gene. Both *UBC-CreERT2*; *Igf1r^fl/fl^* and *Igf1r^fl/fl^* transgenic littermates were treated with tamoxifen at four weeks of age. Cre-ERT2 recombinase expression is induced after tamoxifen administration, and consequent excision of the floxed exon of the Igf1r gene occurs only in *UBC-CreERT2*; *Igf1r^fl/fl^ mice* (gene knockout). The floxed allele remained undeleted in *Igf1r^fl/fl^* (control) mice, but also in some cells of *UBC-CreERT2*; *Igf1r^fl/fl^* animals, where the deletion occurred with different grades of mosaicism depending on tissue and cell types [13]. All experiments and animal procedures were carried out in accordance with the European Communities Council Directive on animal experiments (EU Directive 2010/63/EU, 22 September 2010) and were approved and revised by the Institutional Animal Care and Use Committee (IACUC) from the Center for Biomedical Research of La Rioja, Spain (ref. 03/12).

### 2.3. Sample Preparation and Western Blot

Mouse brains (whole brain minus the cerebellum) were homogenized in ice-cold lysis buffer (25 mM HEPES-Na, 2 mM MgCl_2_, 1 mM EDTA, and 1 mM EGTA) supplemented with 0.1 M phenylmethanesulfonylfluoride (PMSF), 2 mM dithiothreitol (DTT), and a cocktail of proteases and phosphatases inhibitors. The homogenates were incubated on ice for 15 min, frozen and thawed 3 times to favor disruption, centrifuged at 14,000× rpm for 10 min at 4 °C, and the resulting supernatant collected and stored at −80 °C. The amount of protein content in the samples was analyzed by the BCA protein assay using the Pierce™ BCA protein assay kit (Fisher Scientific, Loures, Portugal). For the Western blot, samples were resolved by electrophoresis in 10 and 15% SDS–polyacrylamide gels and transferred to polyvinylidene difluoride (PVDF) membranes. After electrophoresis, proteins were transferred to PVDF membranes, and blocked membranes (1 h in 5% BSA and 0.1% Tween in TBS at room temperature) were incubated overnight at 4 °C with specific antibodies. The proteins were detected separately with specific secondary antibodies. β-actin was used as a loading control. The enhanced chemifluorescent (ECF) detection system (VWR International, Lisboa, Portugal) and Versa Doc or Chemi Doc imaging systems (Bio-Rad, Hercules, CA, USA) were used, and band density was obtained with the Quantity One Software or Image Lab Software, respectively (Bio-Rad, Hercules, CA, USA).

### 2.4. Measurement of Intracellular Reactive Oxygen Species (ROS)

The accumulation of ROS was quantified fluorometrically following the Amplex red–horseradish peroxidase (HRP) method, as described previously [15], with some modifications. Once in the cell, the horseradish peroxidase catalyzes H_2_O_2_-dependent oxidation of non-fluorescent Amplex red into fluorescent resorufin red. Briefly, 50 μg of brain tissue homogenates prepared in ice-lysis buffer without DTT were placed in Krebs buffer and loaded with 50 μM Amplex red reagent and 0.2 U/mL horseradish peroxidase and incubated for 30 min at 37 °C, protected from light. Sample fluorescence corresponding to H_2_O_2_ generation was then monitored for 30 min with 30 s intervals, at 37 °C, with 560 nm excitation and 590 nm emission wavelengths, using a Microplate Spectrofluorometer Gemini EM (Molecular Devices, San Jose, CA, USA). Experiments were performed in duplicate, and Amplex red fluorescence was determined from obtained slopes by subtracting minimal and maximal absorbance values. Further, a control without a sample was performed and subtracted from each value to discard background interference. Amplex red fluorescence was expressed as a percentage of control.

### 2.5. Measurement of Aconitase Activity

Aconitase activity was determined according to Krebs and Holzach [16] and adapted to be spectrophotometrically determined in a SpectraMax Plus 384 microplate reader (Molecular Devices, San Jose, CA, USA). Briefly, brain protein extracts previously homogenated in ice-cold lysis buffer without DTT were diluted in 150 μL buffer containing 50 mM Tris-HCl and 100 mM MnCl_2_ (pH = 7.4), frozen and thawed 2 times and centrifuged for 3 min at 14,000× rpm. Aconitase activity was determined in samples supernatants by monitoring at 240 nm the cis-aconitase after the addition of 20 mM isocitrate, at 25 °C. The activity of aconitase was calculated using a molar coefficient of 3.6 mM^−1^⋅cm^−1^ and expressed as U/mg protein/min. One unit was defined as the amount of enzyme necessary to produce 1 M cis-aconitate per minute.

### 2.6. Measurement of Caspase 3 and Caspase 9-Like Activity

Caspase 3 and caspase 9 activation was measured using a colorimetric method. Briefly, 50 µg of brain tissue homogenate was incubated at 37 °C for 2 h in 25 mM HEPES, pH 7.5 containing 0.1% CHAPS, 10% sucrose, 2 mM DTT, and 40 μM Ac-DEVD-pNA for the determination of caspase 3 and with 40 μM Ac-LEHD-pNA for the determination of caspase 9. The caspase-like activity was determined by measuring substrate cleavage at 405 nm in a microplate spectrophotometer SpectraMax Plus384 (Molecular Devices, San Jose, CA, USA), and values were expressed as a percentage of control.

### 2.7. Measurement of Brain Mitochondrial Enzymatic Activities

All assays described below follow microplate adaptations of previous well-established standard methods for the evaluation of mitochondrial enzyme activities [17,18,19,20]. Of note, for those assays, brain tissue was homogenized in ice-cold lysis buffer without DTT, and mitochondrial respiratory chain complexes activities were corrected and normalized with the citrate synthase activity, an enzyme of the tricarboxylic acid cycle [21].

#### 2.7.1. Mitochondrial Respiratory Chain Complexes Activities

NADH-decylubiquinone oxidoreductase (mitochondrial complex I) activity was determined by a method previously described by Long et al. [18], with some modifications. This assay involves decylubiquinone as the electron acceptor and NADH as a donor. Briefly, 50 μg of total brain homogenate was placed in reaction buffer containing: 25 mM KH_2_PO_4_ (pH 7.5), 5 mM MgCl_2_, 0.3 mM KCN, 4 μM antimycin A, 3 mg/mL BSA, 60 μM coenzyme Q1 and 160 μM DCPIP and baseline readings continuously performed for 3 min with 30 s intervals, at 600 nm, in a SpectraMax Plus 384 microplate reader (Molecular Devices, San Jose, CA, USA). The reaction began with the addition of 100 μM freshly prepared NADH, and readings were continuously recorded for 6 min with 30 s intervals. Finally, 10 μM rotenone (a specific inhibitor of complex I) was added, and reading was continuously performed for 6 min. Enzyme activity was calculated through the mean of slopes obtained during the linear phase and determined as the difference between the activities in the absence and presence of rotenone. A molar extinction coefficient (ε)600 = 19.1 mM^−1^⋅cm^−1^ was applied, and complex I activity was expressed as nmol/min/mg protein.

NADH-cytochrome c oxidoreductase (mitochondrial complexes I + III) activity was determined following the increase in absorbance at 550 nm resulting from the reduction of freshly prepared cytochrome c, as previously described [21]. The assay involves oxidized cytochrome c as an electron acceptor and NADH as an electron donor. Briefly, 50 µg of total brain homogenate was incubated for 2 min in 100 mM phosphate buffer (KH_2_PO_4_; pH 7.4), at 30 °C, followed by the addition of 50 mM Tris-HCl (pH 8.0) medium supplemented with 6 mg/mL BSA, 40 µM oxidized cytochrome c and 240 µM KCN. Baseline readings were performed for 4 min with 30 s intervals at 30 °C, and the reaction was started with the addition of 0.1 mM NADH. Activity reading was continuously performed for 6 min with 30 s intervals at 550 nm in a SpectraMax Plus 384 microplate reader (Molecular Devices, San Jose, CA, USA). Rotenone (4 µM) was added to access the rotenone sensitive activity, and monitoring was performed for 6 additional min. Complexes I + III activity was calculated through the mean of slopes obtained during the linear phase and determined as the difference between basal activity in the absence and presence of rotenone. A ε550 = 19.6 mM^−1^⋅cm^−1^ was applied, and enzymatic activity was expressed as nmol/min/mg protein.

Succinate-ubiquinol cytochrome c oxidoreductase (mitochondrial complexes II + III) activity was determined following the increase in absorbance at 550 nm resulting from the reduction of freshly prepared cytochrome c, the electrons acceptor, and succinate, the electrons donor, as reported elsewhere [21]. Briefly, 50 µg of total brain homogenate was pre-incubated for 5 min, at 37 °C, in 100 mM phosphate reaction buffer (pH 7.4), 1 mM KCN and 500 mM succinate. Reading was initiated by adding the pre-incubated mixture to microplates wells containing 100 mM phosphate reaction buffer (pH 7.4), 0.3 mM EDTA and 100 µM oxidized cytochrome c, and continuous activity was measured for 5 min with 30 s intervals at 37 °C. Antimycin A (40 µM) was then added, and reading was continuously measured for 5 min with 30 s intervals, at 37 °C, at 550 nm in a SpectraMax Plus 384 microplate reader (Molecular Devices, San Jose, CA, USA). Complexes II + III activity was calculated through the mean of slopes obtained during the linear phase and determined as the difference between basal activity in the absence and presence of antimycin A. A ε550 = 19.2 mM^−1^⋅cm^−1^ was applied, and enzymatic activity was expressed as nmol/min/mg protein.

Ubiquinol cytochrome c oxidoreductase (mitochondrial complex III) activity was assayed following the increase in absorbance at 550 nm resulting from the reduction of cytochrome c [21]. The protocol involves oxidized cytochrome c, the electrons acceptor, and decylubiquinol, the electrons donor. Briefly, 50 µg of total brain homogenate was placed in a reaction buffer containing 25 mM KH_2_PO_4_ (pH 7.5), 3.75 µM rotenone, 0.025% Tween-20 and 0.20 mM decylubiquinol. Basal reading was performed for 3 min with 30 s intervals, at 37 °C, in a SpectraMax Plus 384 microplate reader (Molecular Devices, San Jose, CA, USA). Enzymatic activity was initiated with the addition of 75 µM oxidized cytochrome c, and continuous reading was performed for 3 min with 30 s intervals. Finally, 2.5 µM antimycin A was added, and monitoring was performed for 3 additional min. Complex III activity was calculated through the mean of slopes obtained during the linear phase and determined as the difference between basal activity in the absence and presence of rotenone. A ε550 = 19.2 mM^−1^⋅cm^−1^ was applied, and enzymatic activity was expressed as nmol/min/mg protein.

Cytochrome c oxidase (mitochondrial complex IV) activity was determined by a method previously described [22] with some modifications. Briefly, 50 μg of total brain homogenate was incubated, at 37 °C, in reaction buffer containing 50 mM KH_2_PO_4_ (pH 7.0), 4 μM antimycin A and 0.05% n-dodecyl-β-d-maltoside and reading was performed for 3 min with 30 s intervals in a SpectraMax Plus 384 microplate reader (Molecular Devices, San Jose, CA, USA). Enzymatic activity was followed by a decrease in absorbance of reduced cytochrome c at 550 nm, upon addition of 7.2 μM of freshly prepared reduced cytochrome c for 6 min with 30 s intervals. Complex IV activity was calculated through the mean of slopes obtained during the linear phase and determined as the difference between basal activity in the absence and presence of 10 mM of KCN (a specific inhibitor of complex IV). A ε550 = 19.1 mM^−1^⋅cm^−1^ was applied, and mitochondrial complex IV activity was expressed as nmol/min/mg protein.

#### 2.7.2. Mitochondrial Citrate Synthase Activity

To evaluate mitochondrial citrate synthase activity, 50 µg of protein was added to 100 mM Tris-HCl (pH 8) reaction buffer containing 0.1% Triton X-100, 200 µM DTNB, 200 µM acetyl-CoA, and baseline was measured at 412 nm for 3 min with 30 s intervals, at 30 °C. The reaction was started with 100 µM oxaloacetate and followed spectrophotometrically in a SpectraMax Plus 384 microplate reader (Molecular Devices, San Jose, CA, USA) for 6 min by measuring the increase in absorbance at 412 nm (=13.6 mM/cm) resulting from the reduction of DTNB. Results were expressed as nmol/min/mg protein, and mitochondrial respiratory chain complexes activities were normalized by dividing the enzymatic activity rates corresponding to each mitochondrial complex respiratory chain enzyme by the rate of citrate synthase activity [21].

### 2.8. Statistical Analysis

Results are presented as mean ± SEM of the indicated number of experiments. After assessing the normality distribution of the groups using GraphPad Prism Software Inc. (La Jolla, CA, USA), data were analyzed by Student’s unpaired *t*-test or by Mann–Whitney U test where indicated. A *p* value < 0.05 was considered significant unless otherwise specified.

## 3. Results

### 3.1. Characterization of UBC-CreERT2; Igf1r^fl/fl^ Mutant Mice

To study the effects of IGF1R deficiency in the postnatal mouse brain, we used the *UBC-CreERT2*; *Igf1r^fl/fl^* and *Igf1r^fl/fl^* mutant mice that were treated with tamoxifen at four weeks of age to induce Igf1r gene deletion in animals with the *UBC-CreERT2*; *Igf1r^fl/fl^* genotype [13]. Brains were dissected from 13-week-old, fasted males of both genotypes. Brain weights of *UBC-CreERT2*; *Igf1r^fl/fl^* mutant mice (*n* = 14) were significantly lower (89.63%) than those of *Igf1r^fl/fl^* controls (*n* = 11), despite total mouse body weights and the ratio brain/body weight, were not found significantly different at this age (Appendix A). We next surveyed the protein levels of IGF1R and its highly homologous IR in whole-brain extracts (minus the cerebellum) from mice of both genotypes. As found in western immunoblot analysis, the generalized IGF1R deletion evoked a significant decrease in IGF1R protein levels in brains extracts of *UBC-CreERT2*; *Igf1r^fl/fl^* mutants (~45%) (Figure 1a), thus confirming the validity of this mouse model to study the impact of IGF1R deficiency in the brain. Interestingly, the IGF1R downregulation was accompanied by a significant increase in IR protein levels (~125%) (Figure 1b), which could represent a compensatory mechanism to activate downstream signaling cascades.

### 3.2. IGF1R Deficiency Activates PI3K/AKT and Downregulates ERK1/2 Signaling Pathways

Several lines of evidence have demonstrated that IGF1R mediated actions against various neurotoxic cues involve the PI3K/Akt and MAPK/ERK1/2 signaling pathways [23,24]. To determine whether decreased IGF-1R levels compromise signaling in brain tissue, we started by performing immunoblotting densitometric analysis of the catalytic subunit of PI3K, the P13K (p110) protein, and its downstream mediators. We found a significant increase of P13K (p110) protein levels (Figure 2a) as well as of the phosphorylation levels of intracellular serine/threonine kinase Akt, which transduces IGF1R signaling, in brain tissue from *UBC-CreERT2*; *Igf1r^fl/fl^* mutants (Figure 2b). On the other hand, a significant decrease in the pERK/ERK ratio was found in brain tissue from *UBC-CreERT2*; *Igf1r^fl/fl^* mutants (Figure 2c). Altogether these results suggest that IGF1R deficiency impacts the canonical distal pathways that, in this particular mice model, are accompanied by an increase in IR protein levels, representing a compensatory mechanism to promote the activation of the pro-survival pathway PI3K/Akt.

### 3.3. IGF1R Deficiency Evokes a Dual Regulation of Brain GSKβ Activity and Increases Tau Phosphorylation

Akt kinase activation is known to have a strong antiapoptotic action and be directly involved in the phosphorylation of many substrates, including glycogen synthase kinase-3β (GSK3β) at Ser9 [25,26], inhibiting its activity and inhibition of caspase activity, leading to cell survival. According to that, our data show that concomitant with Akt activation, the brain tissue of IGF1R KO mice presented a significant increase in pGSK3β (Ser9) (Figure 3a) and no alterations in caspase-3 and caspase-9-like activities (Figure 3d,e, respectively). These observations suggest that IGF1R deficiency activates PI3K/Akt activation inhibiting GSK3β, probably to preserve brain cells homeostasis and survival.

GSK3β is a multifunctional protein kinase particularly abundant in the brain where it is involved in the regulation of several critical processes, including cell proliferation, cell survival, gene expression, cellular architecture, neural development and plasticity [27]. In particular, GSK3β activation, due to reduced pGSK3βSer9 [28], but also through increased phosphorylation of GSK3β at Y216 [29], has been strongly implicated in the phosphorylation of Tau microtubule-associated protein at the majority of its serine/threonine sites that cause associated toxicities in Alzheimer’s disease (AD). Here, we found that concomitant with an increased level of pGSK3β (Ser9), the brain tissue of IGF1R deficient mice also displayed a significant increase in the pGSK3β (Y216)/GSK3β ratio (Figure 3b), which represents the active form of the enzyme. To further confirm GSK3β activation, we also performed an immunoblotting analysis of Tau phosphorylation at the Ser396, the most favorable target site of GSK3β. The densitometric analysis of blots revealed a significant increase in pTau protein to total Tau protein ratio (Figure 3c). Of note, a significant decrease in the total form of GSK3β and a significant increase in the total levels of Tau protein normalized with actin protein was also detected in brain tissue of mutant mice (Appendix A). These data suggest that in the brains of IGF1R deficient mice, GSK3β activation is dually regulated and can also occur in a PI3K/Akt-independent manner leading to increased phosphorylation of Tau.

### 3.4. Effects of IGF1R Deficiency on Synaptic Plasticity Markers

It is well-known that IGF1/IGF1R signaling can alter the activity of proteins directly involved in synaptic plasticity [30]. To ascertain the consequences of diminished levels of brain IGF1R protein levels, key pre- and postsynaptic integrity markers were assessed by immunoblotting. Brain tissue from *UBC-CreERT2*; *Igf1r^fl/fl^* mutants present a significant increase in the protein levels of the presynaptic marker SNAP25 (Figure 4a), whereas synaptophysin (Figure 4b) and the postsynaptic marker PSD-95 (Figure 4c) remained statistically unchanged. Notably, a significant decrease was found in the ratio pCREB/CREB (Figure 4d). Activation of CREB is widely implicated in neuronal survival and has long been known to be important for the formation of memories. On the opposite, CREB signaling dysfunction is associated with several neurodegenerative diseases, including AD [31] and brain injury models [32]. These results confirm that IGF1Rs deficiency modulates the expression of key regulators of brain higher functions.

### 3.5. IGF1R Deficiency Promotes the Dysregulation of Brain Oxidative Stress Response

Under normal conditions, cells possess several lines of defense working synergistically to protect them against free radical-induced damage. As outlined in previous studies, IGF1R signaling through PI3K/Akt and ERK activation is shown to have a strong involvement in cellular protection against oxidative stress [33]. In particular, both pathways were shown to act as upstream mediators of nuclear factor erythroid 2-related factor 2 activation (NRF2) [34,35], a well-known transcription factor involved in the protection against oxidative damage [36]. Our analysis revealed that NRF2 protein levels were significantly decreased in brain tissue from *UBC-CreERT2*; *Igf1r^fl/fl^* mutants (Figure 5a). We also found a significant decrease in aconitase activity (Figure 5b), which is a key redox sensor of the cells and regarded as a reliable marker of oxidative damage, as well as an increase in H2O2 levels in extracts obtained from the brains of mutant mice (Figure 5c). These observations indicate that IGF1R deficiency evokes a significant increase in brain oxidative stress, supporting its involvement in brain cell redox state.

### 3.6. IGF1R Deficiency Promotes Alterations on Brain Mitochondrial Respiratory Chain Components

Among other aspects of cellular homeostasis, IGF1R signaling is known to have important roles in mitochondrial function [37,38,39]. Following this rationale and the above-mentioned observations, our next task was to decipher the implications of IGF1R deficiency on mitochondrial homeostasis. For that, key markers of mitochondrial biogenesis and mitochondria DNA (mtDNA)-encoded subunits of respiratory chain complexes were assessed by Western blot analysis. It is known that mitochondrial biogenesis is regulated by several transcription factors such as mitochondrial transcription factor A (TFAM), which is responsible for transcription of genes encoded by mtDNA and is essential for mtDNA maintenance, and also by NRF1 and NRF2, both involved in the regulation of the expression of key components of the mitochondrial transcription and translation machinery that are necessary for the production of respiratory subunits encoded by mtDNA [40]. In this respect, we evaluated the protein levels of two important electron transport chain subunits, both encoded by mtDNA, the NADH dehydrogenase subunit 1 (ND1) and the mitochondrial-encoded cytochrome c oxidase 1 (MTCO1), which are subunits of respiratory complex I and IV, respectively. Whereas IGF1R deficiency did not evoke alterations in NRF1 and TFAM protein contents (Figure 6a,b), ND1 expression levels were increased (Figure 6c), and MTCO1 protein levels were decreased (Figure 6d) in brain tissue from *UBC-CreERT2*; *Igf1r^fl/fl^* mutant mice. Considering that ND1 and MTCO1 can be regarded as indirect indicators of activity and quantity of mtDNA [40], our observations suggest that IGF1R brain signaling modulates levels and/or function of mitochondrial respiratory chain subunits, which can contribute to altered mitochondrial respiratory chain activity.

### 3.7. IGF1R Deficiency Affects the Activity of Brain Mitochondrial Respiratory Chain

To verify the functional impact of the alterations found on respiratory chain components, we next sought to evaluate the activity of brain mitochondrial respiratory chain enzymatic complexes. In agreement with the results shown in Figure 6, we found a significant impact of IGF1R deficiency on the activity of brain mitochondrial complexes. In detail, mutant mice present an increased activity of mitochondrial complexes I, I + III (not statistically significant in the last, *p* = 0.0552) and II + III (Figure 7a–c, whereas the activity of complexes III and IV were decreased (Figure 7d,e), although the latter in a non-statistically significant manner (*p* = 0.1121). Citrate synthase activity, used as a quantitative enzyme marker of mitochondrial content and function to normalize mitochondrial respiratory chain complexes activity, was decreased in the brains of *UBC-CreERT2*; *Igf1r^fl/fl^* mutant mice (Appendix A). These observations suggest that the IGF1R deficiency has a strong impact on brain mitochondrial respiratory chain functioning and overall mitochondria activity.

### 3.8. IGF1R Deficiency Decreases Brain Mitochondrial Fusion

Mitochondria are dynamic organelles that undergo continuous fission and fusion, and the balance of these opposing processes regulates mitochondrial morphology. Fission is regulated by dynamin-related protein 1 (DRP1), whereas fusion is regulated by optic atrophy 1 (Opa1) and mitofusin (Mfn) 1 and 2 [41]. Recent studies indicate that IGF1R is able to influence mitochondrial dynamics [38,42]. To further understand the impact of IGF1R deficiency on brain mitochondrial dynamics, brain lysates were used to assess fission and fusion protein levels (Figure 8). Densitometric quantification of the Western blots revealed a significant decrease in the protein levels of Mfn1 (Figure 8b), whereas Mfn2 (Figure 8c), OPA1 (Figure 8a), and activated mitochondrial fission protein pDRP1 remained statistically unchanged (Figure 8d). As described, fission leads to the formation of small, rounded mitochondria, whereas fusion forms extended tubular interconnected mitochondrial networks to allow efficient mixing of mitochondrial content and maintain mitochondrial functionality [41]. These observations suggest that IGF1R deficiency disrupts mitochondrial phenotype towards more fragmented mitochondria, thereby impacting mitochondrial function.

## 4. Discussion

IGF1/IGF1R signaling route is an evolutionarily ancient pathway essential for normal development, growth, and survival [43]. Nevertheless, the observations concerning the involvement of IGF1R signaling and brain function are controversial [44]. Some studies suggest that IGF1/IGF1R signaling is neuroprotective [7,45,46], while others report brain beneficial effects mediated by the genetic reduction of IGF1R levels [47,48,49]. In the present study, by using the recently developed *UBC-CreERT2*; *Igf1r^fl/fl^* mutants, we observed that generalized IGF1R deficiency activates IR/PI3K/Akt and inhibits MAPK/ERK1/2/CREB signaling pathways. Further, GSK3β activation was found to be regulated in a PI3K/Akt-dependent and -independent manner, ultimately leading to an increased Tau phosphorylation, a pathological feature of AD and other tauopathies. IGF1R depleted brains also presented altered mitochondrial respiratory chain enzymatic complexes content and activity, altered mitochondrial dynamics towards a decreased fusion and increased oxidative stress. These observations suggest that the deficiency in IGF1R signaling, despite the tentative compensatory response characterized by the activation of the pro-survival PI3K/Akt branch, is associated with disturbances in mitochondria and redox homeostasis, reinforcing the important role of the crosstalk between IGF1/IGF1R signaling and mitochondria in central nervous system homeostasis.

Generalized deletion of Igf1r mediated by tamoxifen (administrated to 4-week-old mice) diminished IGF1R protein content in brains of adult *UBC-CreERT2*; *Igf1r^fl/fl^* males (13-weeks-old) and caused a significant reduction in brain size of mutant mice without significantly affecting total body weight. Reduced brain size in these mutants has been previously reported when they were analyzed early in development at the age of 8 weeks [13]. In accordance, François et al. [50] have also reported differential diminished brain weight in mutant UBIKOR mice in which ubiquitous IGF1R depletion had been induced by tamoxifen in adulthood. Notably, the authors found that the cognitive and behavioral performance of those mutant mice remained unaffected [50]. In a similar manner, mice with IGF1R suppression, specifically in forebrain neurons, demonstrated a well-preserved brain structure and function, including memory performance, and with neuronal features that even improved, notably synaptic transmission in hippocampal neurons [49]. Consistently, others found that IGF1R knockout neurons presented reduced apical soma and developed leaner dendrites, without alterations in dendrite complexity or axon integrity, important for normal neuronal function [48]. In our animal model of IGF1R deficiency, we detected a distinct pattern of synaptic features characterized by a significant increase in the protein levels of SNAP25 and no significant alterations in the protein levels of synaptophysin and PSD-95. In addition, a significant decrease in CREB activation was found. Through the activity of the downstream effector PI3K/Akt, IGF1R-mediated signaling is described to be involved in synapse development and plasticity [51], in the regulation of PSD95 expression in dendritic spines [52,53] and in the excitatory/inhibitory neurotransmission balance in the brain [54]. Further, both PI3K/Akt and ERK1/2 were shown to target CREB phosphorylation at the Ser133, promoting its activation [55,56]. Looking at our data, it can be suggested that despite the reduced brain size of IFG1R mutants, the compensatory activation of the IR/PI3K/Akt pathway upholds synaptic structure while CREB activation is downregulated due to the decreased activity of the upstream MAP/ERK1/2 pathway. Considering that CREB is of utmost importance for memory formation and consolidation [57], it can be assumed that the decreased ERK1/2/CREB signaling will compromise the cognitive and memory processes of IGF1R mutant mice.

The link between the IGF1/IGF1R axis and brain pathologies, particularly AD, has been the topic of significant interest in recent years, with several studies advocating that the modulation of the IGF1R can alter AD-related pathological features, the deposition of amyloid β-peptide (Aβ) and hyperphosphorylated Tau protein. For instance, ablation and/or inhibition of IGF1R from adult neurons was shown to alleviate AD pathology and associated cognitive deficits in transgenic mouse models of the disease [48,58,59]. Contrariwise, others reported a diminished Aβ accumulation on AD brains after IGF1 treatment, while IGF1R inhibition aggravated both behavioral and pathological AD symptoms in mice [60,61]. Herein, we found that the deficiency of IGF1R caused increased levels of Tau phosphorylation, which was accompanied by an increase in the active form (tyrosine 216 phosphorylation; Y216) as well as an increase in the inactive (serine 9 phosphorylation; Ser9) forms of GSK3β; also called as Tau protein kinase I. Widely known as a downstream substrate of the PI3K/Akt branch, its signaling inactivates GSK3β on Ser9, which becomes unable to phosphorylate the Tau protein [62]. More recently, the active form of the enzyme, which represents half or more of the total GSK3β in cultured cells [63], was also demonstrated to contribute to Tau phosphorylation in neuroblastoma cells exposed to Aβ [29] and to have a contributory role in the pathogenesis of Parkinson’s and Huntington´s diseases by influencing Tau protein phosphorylation [64,65]. Coincident to these observations, we also noticed a significant decrease in the protein levels of total GSK3β, suggesting that the majority of GSK3β was phosphorylated, either in the active or inactive form. This may indicate that GSK3β is dually regulated in the brains of IGF1R KO mice and that pGSK3β(Y216) has a contributory role in Tau phosphorylation in this model, which can further trigger a cascade of deleterious events in brain cells.

Compared to other organs, the brain has high-energy requirements and oxygen consumption. However, it also presents a high content in transition metals and poor antioxidant defenses, being highly vulnerable to damage under conditions of metabolic and oxidative stresses [66]. To counteract these drawbacks, compelling findings disclose an important role of IGF1R/IR intracellular signaling pathways as mediators of brain protection against oxidative stress in diverse injury contexts [67,68,69]. Effects that seem to include reduction of caspases 3 and 9 activation, expression of uncoupling protein 3, stimulation of transcription factors involved in the regulation of antioxidant enzymes (e.g., NRF2 and FOXO) and modulation of mitochondrial functions [37,41,70,71]. In close agreement, our study shows that reduced IGF1R protein levels had a significant impact on brain oxidative status by increasing H_2_O_2_ levels as well as by decreasing NRF2 protein expression and aconitase activity, both known as vulnerable targets of oxidative stress [34,35]. Noteworthy, we also found major effects on brain mitochondrial respiratory chain components and activity in IGF1R deficient mice. As observed, IGF1R deficient mice have increased expression of mt-ND1, a protein component of mitochondrial complex I, in parallel to higher activities of mitochondrial complexes I, I + III (*p* = 0.0552) and II + III. Surprisingly, we also noticed a diminished expression of mitochondrial complex IV MTCO1 component, in parallel with a tendency to reduced activity of complex IV (*p* = 0.1121). As the entry point for most electrons into the mitochondria respiratory chain, complex I play a central role in oxidative phosphorylation and has been suggested as the rate-limiting step in overall electron transfer [72]. Therefore, it seems that in brain tissue of IGF1R deficient mice, mitochondrial respiration and electron transfer are uncoupled, i.e., occur at a higher rate through complex I to complexes II +III, but interrupted at complex III not proceeding to complex IV. Ultimately, this will be transduced in a decreased mitochondrial respiratory function and altered phosphorylation system activity. In a similar manner, the loss of IGF1R in cultured astrocytes altered mitochondrial complex I activity and increased mitochondrial ROS production inducing sensitivity to H_2_O_2_ and cytotoxicity, which resulted in an impaired ability to provide support and protection for neurons under conditions of stress and an impaired metabolism [73,74].

Under normal physiological conditions, mitochondria function and bioenergetics are tightly regulated by structural rearrangements of the organelle, including the remodeling of cristae morphology and elongation or fragmentation of the tubular network organization. For a review, see ref [75]. In this process, mitochondrial fusion is crucial for the maintenance of respiratory capacity in tissues that have high metabolic activity, like the brain [76]. Herein, we show that along with a dysfunctional activity and content of the mitochondrial respiratory chain, brain mitochondria of IGF1R KO mice have an impaired expression of the Mfn1, a critical protein for the mitochondrial fusion process. Such alterations would result in less efficient mitochondria in the brain tissue of the IGF1R KO mice. In close agreement, Pyakurel et al. [77] revealed that Mfn1 is specifically phosphorylated by the MAP/ERK pathway, thereby modulating its participation in apoptosis and mitochondrial fusion in primary cortical neurons. More recently, suppression of IGF1R signaling was found to affect mitochondria dynamics, causing the accumulation of elongated mitochondria in cancer cell lines [38]. These core findings confirm mitochondria as a key target of IGF1R signaling that is likely to influence responses to therapeutic strategies directed to IGF1R.

Noteworthy, although our data demonstrate that a significant reduction of IGF1R expression in *UBC-CreERT2*; *Igf1rfl/fl* mice brain perturbs this organ, the absence of the protein may impact the brain differently. It was previously reported that in this mice model, the activity of Cre to delete exon 3 in the IGF1R gene occurs with different levels of mosaicism in different cell types and seems to vary among tissues/organs [13]. For instance, by Western blot analysis has been observed a 10–20% reduction of IGF1R protein levels in the lungs [14,78], whereas, in adipose tissue and liver, that reduction was 60–80% and 65%, respectively (unpublished data). Further, the accessibility of tamoxifen to different organs and its metabolism in the mouse brain is described to be age, strain- and dose-dependent, which can affect the tamoxifen-dependent recombination rate and the success of the gene silencing [79]. Furthermore, our study does not clarify in which type of brain cells the deficiency of IGF1R occurred. Future studies should focus on the exact localization of the IGF1R in the different cells of the brain, which will give more detailed information about the role of IGF1R in specific brain cells and regions. Nevertheless, independently of the local and brain cells in which IGF1R is reduced, it must be taken into consideration that there is a crosstalk between different brain cells and, independently of the cell type that is more affected, brain alterations described in this paper represents the result of brain cells interaction and adaptation.

## 5. Conclusions

In summary, we have confirmed the importance and complexity of IGF1R signaling in the regulation of brain homeostasis. Our data show that canonical signaling pathways are distinctively modulated and that IRs can, in part, compensate for the absence of IGF1Rs and activate a pro-survival response that may prevent caspases activation but be insufficient to abrogate oxidative stress. Significant disturbances in synaptic integrity and plasticity markers and an increased Tau protein phosphorylation were also observed in the brains with deficient levels of IGF1R. In addition, mitochondria were found to be strongly affected in this animal model showing a dysfunctional respiratory chain activity, impaired expression of proteins involved in mitochondrial dynamics and an altered mitochondrial content.

Collectively, these data suggest that IGF1/IGF1R signaling is crucial for brain homeostasis and, for this reason, interventions that affect IGF1R must be carefully considered to avoid brain dysfunction.

## Figures and Tables

**Figure 1 biomedicines-09-00158-f001:**
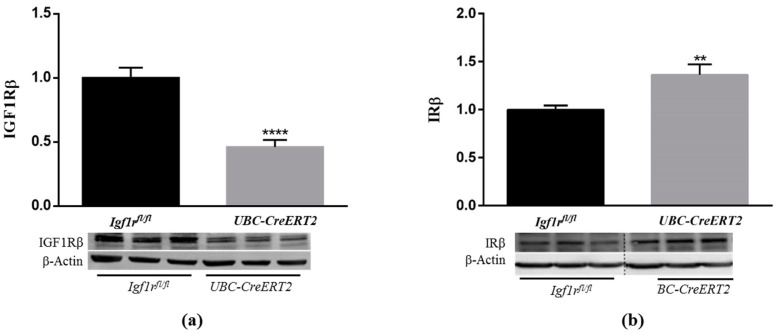
Effects of Igf1r gene deficiency on brain insulin-like growth factor 1 receptor (IGF1R) and insulin receptor (IR) protein levels. (**a**) Western blot analyses of IGF1Rβ protein levels. (**b**) Western blot analyses of IRβ protein levels. Bars in graphs represent the mean ± SEM of the quantification of samples from 8 mice per genotype obtained from each protein band density (upper blots) and normalized with β-actin levels (lower blots) with respect to control mice. Representative immunoblots for β-actin are shown as the loading control. Dotted lines on Western blot images symbolize some removed interspacing lanes for a side-by-side display of samples from both groups. ** *p* < 0.01 and **** *p* < 0.0001 were considered as statistically significant differences. *Igf1r^fl/fl^*—control mice; *UBC-CreERT2*—mutant mice.

**Figure 2 biomedicines-09-00158-f002:**
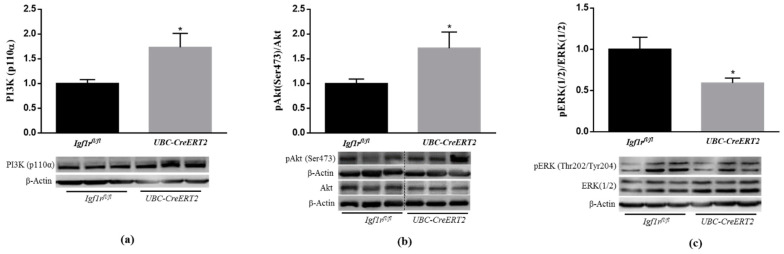
Effects of IGF1R deficiency on brain IGF1R-mediated downstream signaling pathways. (**a**) Western blot analyses of P13K (p110α) protein levels after normalization with β-actin. (**b**) Western blot analyses of the ratio between phosphorylated and total levels of Protein kinase B/Akt (pAkt/total Akt) and (**c**) phosphorylated and total levels of extracellular-signal-regulated kinase (pERK/total ERK). Graphs show mean ± SEM of the quantification of samples from 6 (**a**,**c**) and 8 (**b**) mice per genotype normalized to control mice. Representative immunoblots with β-actin as loading control are shown below the graphs. Dotted lines on Western blot images symbolize some removed interspacing lanes for a side-by-side display of samples from both groups. A * *p* < 0.05 was considered as a statistically significant difference. *Igf1r^fl/f^*—control mice; *UBC-CreERT2*—mutant mice.

**Figure 3 biomedicines-09-00158-f003:**
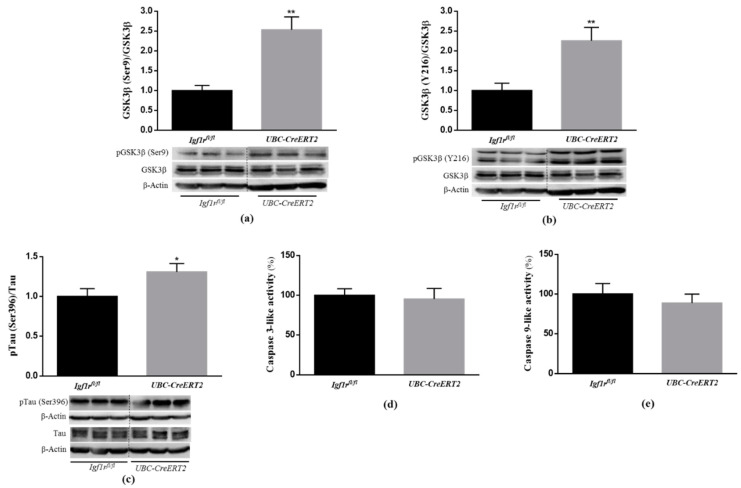
Effects of *Igf1r* gene deficiency on downstream targets of PI3K/Akt signaling. (**a**) Western blot analyses of the ratio between pGSK3β(Ser9)/total GSK3β, (**b**) pGSK3β(Y216)/total GSK3β and (**c**) pTau (Ser396)/totalTau protein levels. (**d**) Caspase-3 activity was determined using the fluorogenic substrate Ac-DEVD-pNA. Assay was performed in duplicate and expressed as percentage of control. (**e**) Caspase-9 activity was determined using the fluorogenic substrate Ac-LEHD-pNA. Assay was performed in duplicate and expressed as percentage of control. Graphs show mean ± SEM of the quantification of samples from 6–10 mice per genotype normalized to control mice. When apply, representative immunoblots are shown below the graph. Dotted lines on Western blot images symbolize some removed interspacing lanes for a side-by-side display of samples from both groups. * *p* < 0.05; ** *p* < 0.01 were considered as statistically significant differences. *Igf1r^fl/fl^*—control mice; *UBC-CreERT2*—mutant mice.

**Figure 4 biomedicines-09-00158-f004:**
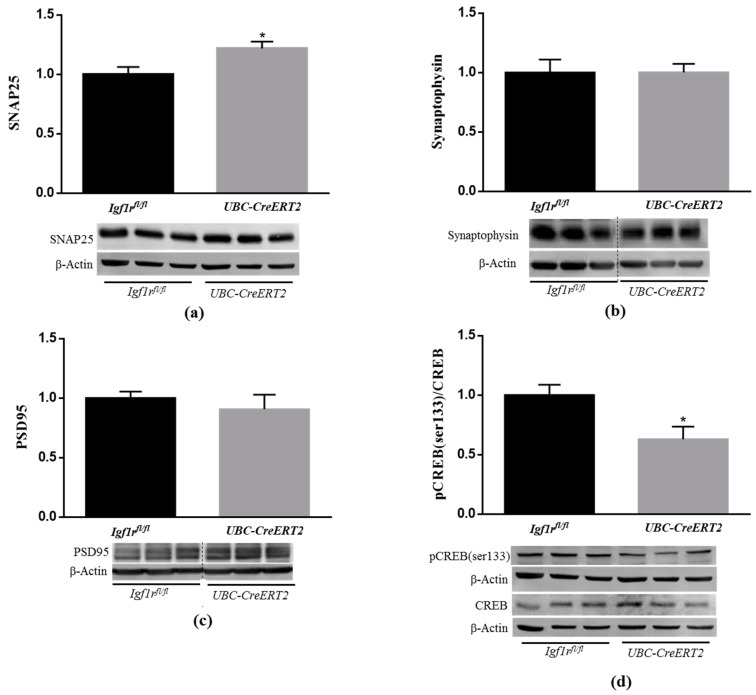
Effects of IGF1R deficiency on markers of neuronal plasticity and synaptic integrity. (**a**) Western blot analyses on SNAP25, (**b**) synaptophysin, (**c**) postsynaptic density protein 95 (PSD95) and (**d**) cAMP response element-binding (CREB) protein levels. Bars in graphs represent the mean ± SEM of the quantification of samples from 9–10 mice per genotype normalized to control mice. Representative immunoblots with β-actin as loading control are shown below the graphs. Dotted lines on Western blot images symbolize some removed interspacing lanes for a side-by-side display of samples from both groups. A * *p* < 0.05 was considered as a statistically significant difference. *Igf1r^fl/fl^*—control mice; *UBC-CreERT2l*—mutant mice.

**Figure 5 biomedicines-09-00158-f005:**
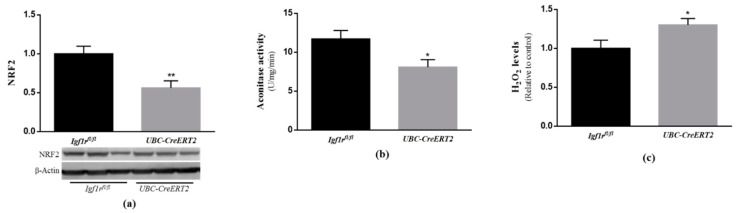
Effects of IGF1R deficiency on brain oxidative status. (**a**) Western blot analyses on nuclear factor E2-related factor 2 (NRF2) protein levels. A representative immunoblot with β-actin as loading control is shown below the graph. (**b**) Aconitase activity, (**c**) H_2_O_2_ levels determined using the Amplex red–horseradish peroxidase (HRP) method. Bars in graphs represent the mean ± SEM values of samples from 6 mice per genotype. Values in A and C are normalized with respect to control mice. * *p* < 0.05; ** *p* < 0.01 were considered as statistically significant differences. *Igf1r^fl/^*—control mice; *UBC-CreERT2*—mutant mice.

**Figure 6 biomedicines-09-00158-f006:**
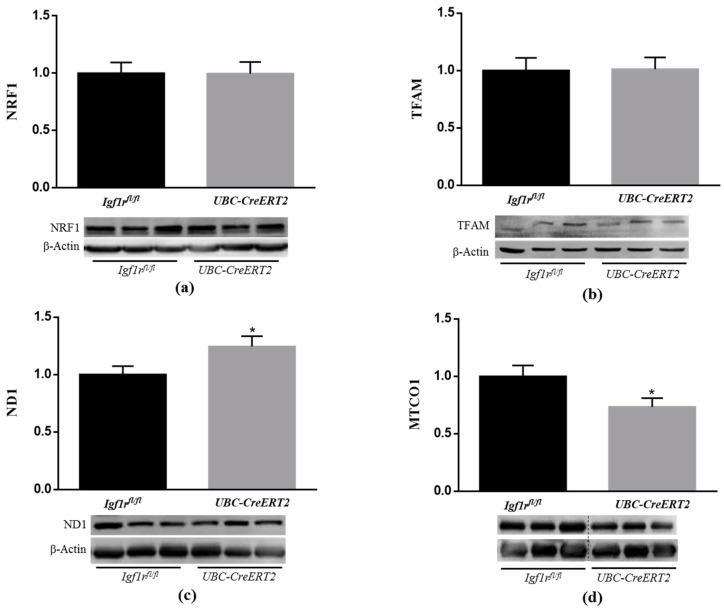
Effects of IGF1R deficiency on mitochondrial biogenesis and protein levels of brain mitochondrial genes. (**a**) Western blot analyses of nuclear factor E2-related factor 1 (NRF1), (**b**) mitochondrial transcription factor A (TFAM), (**c**) NADH dehydrogenase subunit 1 (ND1) and (**d**) mitochondrial-encoded cytochrome c oxidase 1 (MTCO1) protein levels. Bars represent the mean ± SEM of the quantification of samples from 6–9 mice per genotype after normalization with β-actin levels and respect to *Igf1r^fl/fl^* control mice. Representative immunoblots for each protein and their respective β-actin results are shown below the graphs. Dotted lines on Western blot images symbolize some removed interspacing lanes for a side-by-side display of samples from both groups. A * *p* < 0.05 was considered as a statistically significant difference. *Igf1r^fl/fl^*—control mice; *UBC-CreERT2*—mutant mice.

**Figure 7 biomedicines-09-00158-f007:**
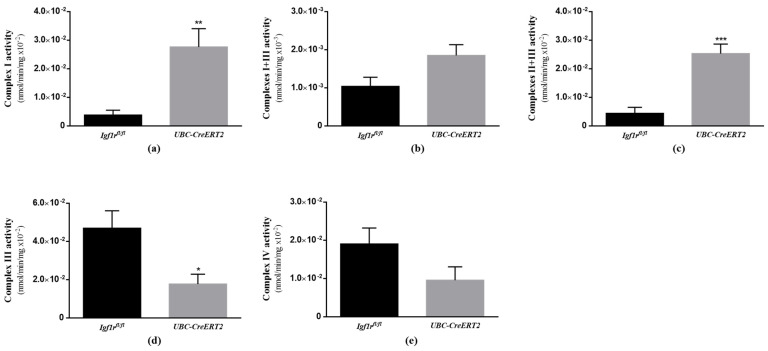
Effects of IGF1R deficiency on the activity of brain mitochondria respiratory chain complexes. (**a**) Mitochondrial complex I activity (*n* = 5 animals per genotype). (**b**) Mitochondrial complex I + III activity (*n* = 5–6 mice per genotype). (**c**) Mitochondrial complex II + III activity (*n* = 6 mice per genotype). (**d**) Mitochondrial complex III activity (*n* = 5–6 mice per genotype). (**e**) Mitochondrial complex IV activity (*n* = 5–6 mice per genotype). Bars in graphs represent the mean ± SEM normalized to citrate synthase activity. * *p* < 0.05; ** *p* < 0.01; *** *p* < 0.001 were considered as statistically significant differences. *Igf1r^fl/fl^*—control mice; *UBC-CreERT2*—mutant mice.

**Figure 8 biomedicines-09-00158-f008:**
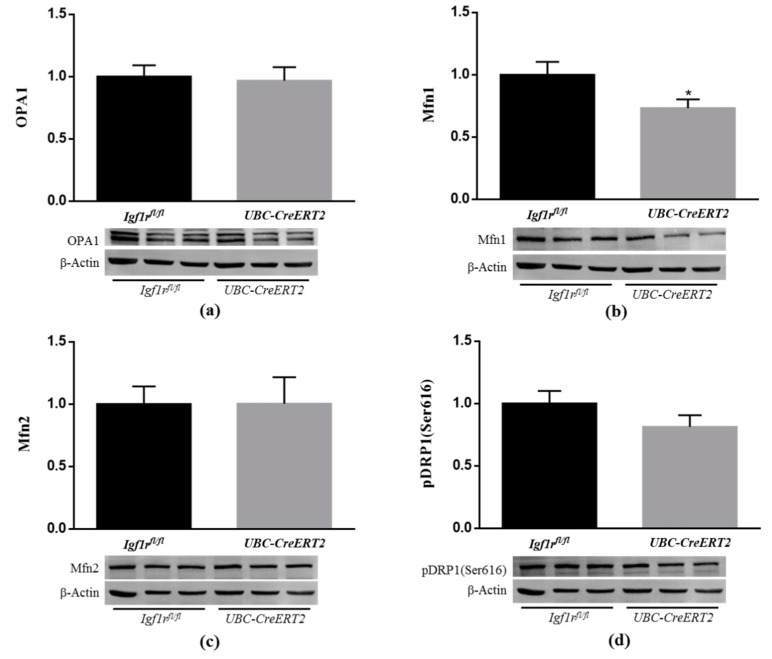
Effects of IGF1R deficiency on brain mitochondria dynamics. (**a**) Western blot analyses of OPA1, (**b**) Mfn1, (**c**) Mfn2 and (**d**) pDRP1 (Ser616) protein levels. Bars represent the mean ± SEM of the quantification of samples from 8 mice per genotype after normalization with β-actin levels and respect to *Igf1r^fl/fl^* control mice. Representative immunoblots for each protein and their respective β-actin results are shown below the graphs. A * *p* < 0.05 was considered as a statistically significant difference. *Igf1r^fl/fl^*—control mice; *UBC-CreERT2*—mutant mice.

## Data Availability

Not applicable.

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
