# Peer review of "IGF1R Deficiency Modulates Brain Signaling Pathways and Disturbs Mitochondria and Redox Homeostasis"

_biomedicines, 2021, doi:10.3390/biomedicines9020158_

Round 1

Reviewer 1 Report

The manuscript by Cardoso et al., describes the Insulin-like growth factor 1 receptor (IGF1R) deficiency and the impact on brain signaling pathways and mitochondrial homeostasis.  The proposal uses male mice with or without the IGF1R deficient mice as directed by the Cre-ERT2 recombinase expression via tamoxifen treatment.  The IGF1R is decreased in most tissues and analysis of brain expression and function was determined.  KO of IGF1R in the brain resulted in a 50% decrease in protein levels and a slight but significant increase in the insulin receptor.  Changes in other parameters are detected and reported.   The overall conclusions are that IGF1R signaling is important for brain homeostasis and is critical for caspase activation, ROS reduction, etc.  Several issues need addressing.  

1) It is hard to really comprehend the importance of the IGF1R in the brain and homeostasis without specific localization of the IGF1R in the different cells of the brain.  Why only a 50% reduction?  Where is it reduced, neuronal, glia, astrocytes, blood vessels?  This needs to be determined.

2) Same with IRbeta, where is it elevated?

3) Are the signaling pathways altered in which cells?  Probably isolated neurons and glia would help with a better understanding of the role of IGF1R in the brain.  I.e., isolated cells from the IGF1R deleted animals and examine in vitro.

4) pTau is only one component of the AD pathway.  Phosphorylation alone is not sufficient.

5) Some of the changes are modest and might not have that significant a role in brain homeostasis.  

Author Response

Answers to Reviewers comments and suggestions:

Reviewer #1:

Comments to the Author:

The manuscript by Cardoso et al., describes the Insulin-like growth factor 1 receptor (IGF1R) deficiency and the impact on brain signaling pathways and mitochondrial homeostasis.  The proposal uses male mice with or without the IGF1R deficient mice as directed by the Cre-ERT2 recombinase expression via tamoxifen treatment.  The IGF1R is decreased in most tissues and analysis of brain expression and function was determined.  KO of IGF1R in the brain resulted in a 50% decrease in protein levels and a slight but significant increase in the insulin receptor.  Changes in other parameters are detected and reported.   The overall conclusions are that IGF1R signaling is important for brain homeostasis and is critical for caspase activation, ROS reduction, etc.  Several issues need addressing. 

1) It is hard to really comprehend the importance of the IGF1R in the brain and homeostasis without specific localization of the IGF1R in the different cells of the brain.  Why only a 50% reduction?  Where is it reduced, neuronal, glia, astrocytes, blood vessels?  This needs to be determined.

Answer: We do understand the reviewer concern and are thankful for the pertinent comment.

As highlighted in the last paragraph of the discussion section of the manuscript, we do agree that the model has limitations, and the methodology to determine Cre efficiency at the cellular level is limited too. The activity of Cre in these mice to delete exon 3 in the IGF1R gene is chimeric, and seems to vary among tissues. By western blot analysis we observed that the alterations of IGF1R protein levels varies significantly in different tissues. For instance, in the lung reducing protein amounts to 10-20% (Piñeiro-Hermida et al. Sci Rep 7:4290, 2017;  Piñeiro-Hermida et al. Allergy 72:1317, 2017) whereas it goes down to 60-80% in adipose tissue and 65% in liver of these mice (manuscript in preparation). On the contrary, it seems that it does not work so efficiently in the cochlea (inner ear). These variability on Cre efficiency may depend on tissues and cell types, as previously discussed (Lopez et al. Trangenic Res.24:279, 2015). In addition, accessibility and metabolism of tamoxifen to different organs may be an additional issue. As shown, tamoxifen metabolism in mouse brain is age-, strain- and dose-dependent, which can affect the tamoxifen-dependent recombination rate (Valny et al. Front Cell Neurosci. 10: 243, 2016). So, those reasons can explain why we don´t have a full reduction of IGF1R protein levels in brain tissue.

Regarding the determination of the specific localization of the IGF1Rs in the brain, please note that the present study was performed in brain tissue homogenates, and that specific evaluation would only be possible with immunohistochemistry. However, this would require more animals to perfuse and section and presently, we cannot do that. Further, previous experience tells us that there are no anti-IGF1R antibodies that stain specifically IGFR in mouse brain sections, particularly due to the cross-reaction the Insulin receptor.

We must also take into consideration that there is a crosstalk between different brain cells and, independently of the cell type that is more affected, brain alterations described in this paper represents the result of brain cells interaction and adaptation.

Nevertheless, independently of the local and brain cells in which IGF1R is reduced, data obtained in the present manuscript reveals that even 50% of reduction in IGF1R protein levels is enough to evoke significant effects in overall brain tissue homeostasis and mitochondrial function, confirming the complexity of IGF1R in mediating several brain responses.

2) Same with IRbeta, where is it elevated?

Answer: As with IGF1R, it is not possible to assess the exact localization of IRs because the study was performed in brain tissue homogenates. That would require a new set of animals to perfuse and perform immunohistochemistry, and again there are no good specific InsR antibodies to stain sectioned brain tissue. It is a big technical problem in the field.

3) Are the signaling pathways altered in which cells?  Probably isolated neurons and glia would help with a better understanding of the role of IGF1R in the brain.  I.e., isolated cells from the IGF1R deleted animals and examine in vitro.

Answer: We do hope that the reviewer understands but it is not possible for us at the time to have a new set of KO animals and perform those additional experiments. That would require several months to obtain reliable results (the financial costs associated to the suggested experiments together with the pandemic COVID-19 situation impede us from doing new studies).

So, as with the above comments, even though we agree with the reviewer that the information attained with those additional experiments would give important information about the role of IGF1R in specific brain regions and cells, our data gives an important input to the field; it not only elucidates the effects of IGF1R deficiency in a physiological context, i.e. without a disease, but also highlights the need to consider those aspects when designing experimental interventions blocking IGF1/IGF1R signaling in a context of disease. And this is a situation that is not often considered.

4) pTau is only one component of the AD pathway.  Phosphorylation alone is not sufficient.

Answer: We do agree with the reviewer. We sought to determine pTau levels since this protein is a downstream target of the IR/IGF1R signaling pathways; its evaluation as well as the activation status of the GSK3beta, the direct kinase responsible for Tau phosphorylation, could give us further information about the physiological absence/reduction of IGF1Rs in brain tissue.

5) Some of the changes are modest and might not have that significant a role in brain homeostasis. 

Answer: In our opinion, even though some changes in signaling mediators’ may be modest, data shows that there is a crosstalk between the different IGF1R mediators and/or effectors in brain cells to upkeep brain tissue homeostasis in the absence/reduction of IFG1R protein levels. 

Reviewer 2 Report

The manuscript reported characterization of the PI3K/Akt and MAPK/ERK1/2 signaling pathways and mitochondria-related parameters in IGF1R deficient mouse brain. Although that the PI3K/Akt and MAPK/ERK1/2 are impaired in IGF1R deficient cells is not new, the authors focused the investigation specifically on the brain tissues, and in addition to the pathways, synaptic integrity, caspases 3 and 9-like activity, mitochondrial dynamics, content and respiratory chain complexes activity, and general brain oxidative status were also explored in this study, which provided some fundamental data for future investigation. The experiment was well designed, and the manuscript was well written. While there is a major concern regarding the WB images showed in the manuscript.

  • In Figure 2c, two bands were shown when probing with pERK and ERK antibodies, so, which band was used to compare and why? Same concern in Figure 3b and Figure 8a.
  • Although the mice number was mentioned for each WB, there was no description of the numbers of experiments for each WB in all figures. Were those results consistent if the WBs were run in triplicate as routine?
  • The WB image selected to show in the manuscript should be representative and consistent with the quantification. While the WBs in Fig1b, Fig4b and c, and Fig6b and their quantification do not seem convincing. It would be nice to replace those images with more convictive representative images.
  • Ideally, although not necessary for the publication of this study, the mRNA level should be shown to support the WB, which can make the conclusion much stronger. Otherwise, the WBs for other mice (other than the three mice for each genotype showed in each figure) should be shown as supplementary data.

Author Response

Reviewer #2:

Comments to the Author:

The manuscript reported characterization of the PI3K/Akt and MAPK/ERK1/2 signaling pathways and mitochondria-related parameters in IGF1R deficient mouse brain. Although that the PI3K/Akt and MAPK/ERK1/2 are impaired in IGF1R deficient cells is not new, the authors focused the investigation specifically on the brain tissues, and in addition to the pathways, synaptic integrity, caspases 3 and 9-like activity, mitochondrial dynamics, content and respiratory chain complexes activity, and general brain oxidative status were also explored in this study, which provided some fundamental data for future investigation. The experiment was well designed, and the manuscript was well written. While there is a major concern regarding the WB images showed in the manuscript.

1) In Figure 2c, two bands were shown when probing with pERK and ERK antibodies, so, which band was used to compare and why? Same concern in Figure 3b and Figure 8a.

Answer: The quantification of both pERK and ERK was made considering the two bands. As specified in the antibodies suppliers (which are indicated in the materials and methods section of the manuscript), mitogen-activated protein kinase (MAPK) signaling pathways involve two closely related MAP kinases, known as extracellular-signal-related kinase 1 (ERK 1, p44) and 2 (ERK 2, p42). The used antibodies recognize the two bands that must be quantified together.

As with OPA1, the two bands observed are the two isoforms of the protein and this is the common pattern of detection, also shown by others. So, and accordingly to the antibody supplier (indicated in the materials and methods section of the manuscript), the two bands were quantified together to determine the expression of the protein.

The same goes with Figure 3b, the two bands were used to quantify the expression of pGSK3β(Y216).

2) Although the mice number was mentioned for each WB, there was no description of the numbers of experiments for each WB in all figures. Were those results consistent if the WBs were run in triplicate as routine?

Answer: Since the study was performed in brain tissue, each lane of the WBs represents one animal of respective genotype. The WBs were not run in triplicate and the data obtained from the mice number mentioned in each WB was used to determine the mean ± SEM of the quantification of studied proteins.

3) The WB image selected to show in the manuscript should be representative and consistent with the quantification. While the WBs in Fig1b, Fig4b and c, and Fig6b and their quantification do not seem convincing. It would be nice to replace those images with more convictive representative images.

Answer: We have replaced those images with new ones. Please see the new representative images of Figure 1b, Figure 4b and c and Figure 6b in the revised manuscript.

4) Ideally, although not necessary for the publication of this study, the mRNA level should be shown to support the WB, which can make the conclusion much stronger. Otherwise, the WBs for other mice (other than the three mice for each genotype showed in each figure) should be shown as supplementary data.

Answer: We thank the reviewer for the suggestion but, we do not have more material to perform the mRNA determination. As suggested, we have uploaded the uncropped WBs images of all blots analyzed as supplementary data. Please see the section “Supplementary data” of the revised manuscript.

Round 2

Reviewer 2 Report

The concerns from this reviewer have been addressed.

Author Response

Thank you for accepting our responses.